# Sterile but Sexy: Assessing the Mating Competitiveness of Irradiated *Bagrada hilaris* Males for the Development of a Sterile Insect Technique

**DOI:** 10.3390/insects16040391

**Published:** 2025-04-07

**Authors:** Chiara Elvira Mainardi, Chiara Peccerillo, Sergio Musmeci, Alessandra Paolini, René F. H. Sforza, Alessia Cemmi, Ilaria Di Sarcina, Gianfranco Anfora, Daniele Porretta, Francesca Marini, Massimo Cristofaro

**Affiliations:** 1Biotechnology and Biological Control Agency (BBCA) Onlus, Via Angelo Signorelli 105, 00123 Rome, Italyfra.rini.bbca@gmail.com (F.M.); m.cristofaro55@gmail.com (M.C.); 2Department of Environmental Biology, University of Rome “La Sapienza”, 00185 Rome, Italy; daniele.porretta@uniroma1.it; 3Center of Agriculture, Food and Environment (C3A), University of Trento, 38010 San Michele all’Adige, Italy; gianfranco.anfora@unitn.it; 4ENEA, Casaccia Research Center, SSPT-AGROS-Agriculture 4.0 Laboratory, Via Anguillarese 301, 00123 Rome, Italy; sergio.musmeci@enea.it; 5United States Department of Agriculture, European Biological Control Laboratory (USDA-ARS-EBCL), 810 Avenue du Campus Agropolis, 34980 Montferrier-sur-Lez, France; rsforza@ars-ebcl.org; 6ENEA, Casaccia Research Center, NUC-IRAD-GAM Laboratory, Via Anguillarese 301, 00123 Rome, Italy; alessia.cemmi@enea.it (A.C.); ilaria.disarcina@enea.it (I.D.S.)

**Keywords:** sterile insect technique, gamma irradiation, biological control, insect pest, stink bug

## Abstract

Invasive agricultural insects cause severe economic losses worldwide due to their feeding activity on crops. The painted bug *Bagrada hilaris* (Hemiptera: Pentatomidae) is currently controlled almost exclusively through the spraying of synthetic insecticides. Among the possible least toxic alternative approaches, one possible solution is the Sterile Insect Technique (SIT), which involves sterilizing males by gamma irradiation and releasing them into the field so that they mate with wild females, thereby reducing the sustainability of the target population. In this study, we analyzed the sexual behavior of irradiated and non-irradiated males to test whether sterile males remain competitive with their fertile counterparts. After identifying the optimal irradiation dose, we compared the mating duration of treated males under different conditions. The results showed that irradiation does not affect the sexual performance of males; in fact, they appear to perform better than non-irradiated males. These data indicate that SIT could be an effective tool for controlling *B. hilaris*, helping to reduce pesticide use and promote more sustainable management of this invasive species.

## 1. Introduction

Among the major global changes in the 21st century, anthropogenic drivers have caused an increase in invasive alien species [1]. An insect that has recently expanded its range to become an agricultural pest in many parts of the world is the pentatomid bug *Bagrada hilaris* Burmeister (Hemiptera: Pentatomidae), commonly known as the painted bug. Originally distributed in Africa, the Middle East, and Southern Asia [2,3], it has rapidly become an invasive pest in the last few years in several other areas, representing a serious threat to many economically important crops mainly belonging to the Brassicaceae family [4]. Since it was reported in California in 2008, it has extended its distribution to central Arizona [5], Nevada, New Mexico, and Utah [4]. Recently, it has been recorded in Mexico [6,7], Hawaii [8], Chile [9], and Argentina [10]. In Europe, *B. hilaris* was first identified in the late 1970s confined on the island of Pantelleria (Italy) and Malta [11]; since then, it has established mainly on caper plants (*Capparis spinosa* L.), becoming a key pest of this crop in the island [12].

*Bagrada hilaris* exhibits a “lacerate-and-flush” method of feeding that causes circular chlorotic lesions to the leaves that eventually result in the destruction of apical meristems, thus blocking the growth of the terminal part of the shoot [13,14]. The mechanical feeding damage is amplified by the characteristic gregarious behavior of nymphal instars. In case of severe injuries, affected host plants may be considered unmarketable [15].

Control of populations is generally achieved with conventional chemical insecticides such as pyrethroids and carbamates [16]. In addition to being an expensive and ecologically unsustainable solution, the increasing damage to crops caused by this invasive species is probably due to the development of pesticide resistance [8,16]. Consequently, there is an urgent demand to develop new damage mitigation and prevention strategies more species-specific and with a significantly lower treatment cost [17].

For a possible development of classic and augmentative biological control programs against *B. hilaris*, a very small number of egg-parasitoid species has been identified and taken under consideration [18,19,20]. Moreover, the bagrada bug has a peculiar oviposition behavior, which differs from that of other Pentatomidae: eggs are laid singly or in small groups underneath the leaves, on stems, and/or very often buried under the first 2–4 mm of sandy soils [2,21,22]. This particular behavior of oviposition requires parasitoids coevolved with this species, as *Gryon aetherium* Talamas (Hymenoptera, Scelionidae) turns out [23].

In this scenario, the present work is based on the innovative idea of using the Sterile Insect Technique (SIT) in a context of an area-wide integrated pest management as a possible control strategy to suppress the *B. hilaris* populations alone and/or in combination with other biocontrol methods to suppress the populations of this pest species when conditions of isolation such as on the Pantelleria island (Italy) are met.

The management of a pest through a classic SIT approach involves the mass breeding of the target species, the sterilization of males, and their periodic and controlled release into the environment in order to drastically reduce pest reproduction and reduce the population size, controlling population growth. This technique, therefore, has no effect on off-target organisms and is free from possible resistance mechanisms [24,25,26].

Although SIT has been successfully employed in multiple insect orders (especially Diptera and Lepidoptera) [27], studies involving Hemiptera pentatomid are more recent and limited. One of the assumptions necessary to ensure the success of the technique is that adults do not cause direct damage [26]. The release of sterile adult hemipterans can therefore induce a major loss to the target cultures, which is the reason why the literature available on the biology of irradiated hemipterans is still very scarce and studies available concerned only *Halyomorpha halys* Stål (Hemiptera: Pentatomidae) [28,29,30] and *Nezara viridula* L. (Hemiptera: Pentatomidae) [31,32,33].

However, in particular geographical and/or physical conditions (e.g., greenhouses and isolated geographic areas) and under certain assumptions required by an SIT program, the application of such a control method may be able to contain the pest population to the point of achieving its total eradication [29]. In addition, the major feeding damage is caused by adult females and juvenile instars that exhibit gregarious behavior [13]. However, with the aim of total eradication from an isolated area, even the potential temporary damage caused by the released sterile males could be acceptable and socio-economically sustainable. Based upon these assumptions, considering the biology of the target pest species and the geographical characteristics of the island, Pantelleria could be selected as a feasible model area to evaluate SIT as a suitable strategy for the eradication of *B. hilaris*.

Irradiation by gamma rays of *B. hilaris* adults was first considered as a control method in the study of Cristofaro and colleagues [34]. The study revealed that, with the mating among virgin fertile females and males irradiated at 64 Gy, it is possible to achieve 90% sterility, and that 100 Gy is the lowest dose to approach total sterility.

Although we have information on fertility suppression at different doses of irradiation, it is evident that there is a need to determine the optimal dose of irradiation that will represent a correct trade-off between sterility, longevity, and competitiveness.

For SIT to be successful, irradiated males must retain the ability to interact and mate efficiently with their wild counterparts [26]. Irradiated males must retain the ability to maintain mating propensity, locate mates, engage in copulation, and successfully inseminate, despite the physiological damage caused by radiation [35]. To ensure compatibility between released insects and the target insect population, comprehensive studies on sexual behavior are needed to enable the development of appropriate protocols to ensure the quality of reared individuals.

In cases where females actively select mates based on the male phenotype, the mating behavior of sterile males becomes particularly critical. Even small behavioral differences between wild and sterile males can result in reduced competitiveness [26]. In addition, selection pressures may favor wild females that can identify and reject sterile males, leading to wild populations that are “behaviorally resistant” to SIT [36]. In the family Pentatomidae, a recurrent aspect of sexual behavior concerns the influence of the female’s response to initial contact with the male. This response often plays an active role in partner choice, with females demonstrating a willingness or refusal to mate by facilitating copulation or deliberately avoiding it. In contrast, males appear to be responsible for controlling mating duration [37,38,39].

Therefore, the detailed deciphering of these aspects of the reproductive biology of the target species is an essential prerequisite for assessing potential differences between irradiated and non-irradiated males for the development of the SIT. Our current understanding of the biology and behavior of *B. hilaris* as a function of developing innovative control techniques is limited; to fill this knowledge gap, this study aims to provide a starting point for investigating the mating performance of irradiated males of *B. hilaris*.

The objectives of this study are as follows:(a)Evaluation of the sexual performance between males irradiated at three different doses (60, 80, and 100 Gy) and non-irradiated males (control), in terms of the amount of time taken to initiate the first copulation, as well as the number and duration of matings over the 18 h of the bioassay.(b)Study of sexual behavior at the optimal dose: once the first experiment was completed, we evaluated differences in the mating success of irradiated and non-irradiated males under both no-choice and choice conditions. In addition, differences in feeding activity between irradiated males, non-irradiated males, and females were assessed. Unlike the first bioassay, observations were carried out for three days, allowing us to assess the variation in the performance of the irradiated males over time.

By conducting these investigations, we aim to contribute to a deeper understanding of the mating behavior and reproductive dynamics of *B. hilaris*, under controlled conditions, shedding light on the effectiveness of irradiated males in terms of their mating performance. Furthermore, the assessment of feeding behavior will provide insights into potential differences in nutritional requirements or feeding patterns between male and female individuals. The ultimate goal is to contribute to completing all the fundamental knowledge required for the development of an SIT strategy against this species.

## 2. Materials and Methods

### 2.1. Insect Collection and Rearing

During the late summer and the autumn 2023, individuals of *B. hilaris* (mainly adults, but also late instar nymphs, 4th and 5th) were collected at the “Cooperativa Agricola Produttori Capperi”, on the island of Pantelleria (Italy) (GPS coordinates 36°46′23″ N 11°57′41″ E). The number of specimens collected varied by season, ranging from one hundred to one thousand individuals per sampling event. Collection was carried out using an entomological aspirator. Captures were most successful during the hottest hours, i.e., from late morning until mid-afternoon. Individuals were transported in cardboard tubes (4.1 × 11.9 cm). Several cubic sleeve cloth cages (60 × 60 × 60 cm, Bugdorm, Taichung City, Taiwan), placed in 4 larger cages (120 × 60 × 60 cm, Bugdorm, Taiwan), were set up at the quarantine facility of the Fondazione Edmund Mach, San Michele all’Adige (TN), Italy. Some sheets of paper were placed on the bottom of the cages to maintain the correct percentage of humidity, and three Petri dishes (9 cm Ø), filled with a 3 mm layer of fine sand (granulometry = 200 µm), were provided as oviposition sites. The rearing temperature was set at 25 ± 1 °C, relative humidity was 60%, and the light/dark cycle was 16:8. *Brassica oleracea* L. var. *gemmifera* (Brussels sprouts) was placed as a food source. Each cage was cleaned, and the food was replaced three times a week to allow the insects to always have a fresh food source.

### 2.2. Irradiation Process

Newly emerged fifth-instar nymphs were taken from the laboratory colony by plastic 50 mL CorningTM Falcon tubes. These individuals were then isolated in 5 cm diameter Petri dishes until they reached adulthood. This process allowed us to keep separate and then identify males from females based on sexual dimorphism [8], ensuring their virginity. *B. hilaris* females reach full sexual maturity 1–2 days after emergence [40], whereas males are ready to mate immediately upon eclosion (personal observations). Therefore, isolating the final nymphal instar ensured both the sexual maturity and the preservation of their virgin status.

Groups of 20 newly emerged males were placed in Petri dishes (9 cm Ø) and sent from the Edmund Mach Foundation to Rome (Italy) by DHL using a “5-level” packing procedure. They were irradiated at three different doses (60, 80, and 100 g) at the Calliope gamma Facility of the ENEA Casaccia Research Centre (Rome, Italy) [41], equipped with 25 Cobalt-60 radioisotope sources (average energy 1.25 MeV). The dose rate was 175.03 Gy/h (2.92 Gy/min). Non-irradiated females and males, to be used as controls, were also placed under the same controlled conditions as the irradiated individuals.

At the end of the process, all material was returned to the quarantine facility of the Edmund Mach Foundation, following the same procedure.

### 2.3. Evaluation of Sexual Performance at Three Different Doses

A no-choice behavioral bioassay was performed. Only virgin adults were used in the study of courtship and copulation to reduce possible behavioral variability due to learned sexual behavior. The male individuals treated were exposed to three different doses: 60, 80, and 100 Gy. In a previous study [34], doses ranging from 16 to 140 Gy were tested. When irradiated males were mated with unirradiated females, egg hatch fell to about 10% at 60 Gy, 5% at 80 Gy, and was completely suppressed at 100 Gy. Based on these results, we selected 60, 80, and 100 Gy to evaluate the sexual behavior of irradiated males.

The experiments started the day after the irradiation.

For each treatment, one pair of individuals was placed in a 500 mL transparent glass jar covered with a 680 µm white polyester mesh, with a Petri dish cap (5 cm Ø) filled with fine sand as an oviposition site. Each pair was fed a single Brussels sprout (*Brassica oleracea* var. gemmifera). Bioassays were performed at 28° ± 2 C, 16:8 L:D, RH = 60%, and an illuminance of ≈800 lx. In each jar, there was one non-irradiated virgin female and one irradiated male (either at 60, 80, or 100 Gy). For each dose, 20 replicates were set up. The control consisted of one non-irradiated female and one non-irradiated male (20 replicates). Individuals were observed for 18 consecutive hours, and every 15 min, a note was taken of the presence/absence of mating.

The time interval of 18 consecutive hours was considered the best for this preliminary experiment, because the mating in *B. hilaris* can last for several hours [42]; therefore, to obtain an optimal estimate of the time spent in matings, prolonging the observations over an extended period of time was of paramount importance. The experiments started at 10 a.m. and ended at 4 a.m. This time slot was selected as it included all phases of the day (morning, afternoon, and night). To observe the individuals during the hours of darkness, a red headlamp (GearLight S500 LED, Walpole, MA, USA) was used.

To evaluate the sexual performance of irradiated males, we recorded the following parameters: (i) the time elapsed before the first copulation, (ii) the number of copulation events during observations, (iii) the proportion of time spent in copula (mating rate), and (iv) the duration of the first copulation. These indicators offer a detailed view of how irradiation affects reproductive behavior.

### 2.4. Study of Sexual Behavior at the Optimal Dose

By using an entomological aspirator, nymphs at the fifth instar were singularly transferred from the laboratory breeding into Petri dishes (5 cm Ø) until they reached the adult stage. Then, they were separated according to their sexual dimorphism to keep the virgin physiological status. Groups of 10 males were then placed in Petri dishes (9 cm Ø) and irradiated at 80 Gy at the Calliope facility of the ENEA Casaccia Research Center (Rome, Italy), following the same procedure as in the above experiment. Non-irradiated males (control) were placed under the same controlled conditions as the irradiated individuals; i.e., they made the journey to Rome to avoid confounding factors due to stress.

To conduct the tests under no-choice conditions, a pair of individuals (one non-irradiated female and one male irradiated at 80 Gy) was placed in a 500 mL clear glass jar covered with a 680 µm white polyester mesh. Each pair was fed a single Brussels sprout. The control group consisted of one non-irradiated female and one non-irradiated male. For both the treatment and control, 100 replicates were performed.

The tests were performed at a temperature of 28 ± 2 °C, with a photoperiod of 16:8 (L:D), relative humidity of 60%, and illumination of ~800 lx. The experiment started the day after irradiation. Individuals were observed for three consecutive days from 10 a.m. to 4 p.m., and mating presence/absence and feeding activity were noted every hour. This time slot was chosen because Huang and colleagues [13] found that *B. hilaris* females mate all day long, peaking between 10 a.m. and 4 p.m.

The same experimental design was applied to the tests under choice conditions. Three individuals were isolated from each jar: one non-irradiated female, one non-irradiated male, and one irradiated male (80 Gy). Two types of controls were used: a positive control, in which the female was isolated with two non-irradiated males, and a negative control, in which the female was confined with two irradiated (80 Gy) males. For both treatment and control groups, 100 replicates were performed.

To understand which male the female was mating with, the non-irradiated and irradiated males were marked on the pronotum using water-based acrylic markers (Uniposca, Posca brand, model PC-3M, Mitsubishi Pencil Co., Tokyo, Japan). The choice of this marker was motivated by its nontoxicity and durability. The colors used were yellow and white, respectively. Immediately after irradiation, each male was isolated in a Petri dish (9 cm Ø), and the mark was placed on the pronotum of the insect. Fifty trials were conducted in which the irradiated male was colored yellow and the non-irradiated one white, and another fifty trials were conducted in which the colors were reversed. In the controls, both males were marked to show the possible effect of marking on mating.

Individuals were observed for three consecutive days from 10 a.m. to 4 p.m., and the presence/absence of mating and feeding activity were noted every hour.

### 2.5. Data Analysis

#### 2.5.1. Evaluation of Sexual Performance at Three Different Doses

A preliminary analysis with a no-choice experimental design was performed using several doses of irradiation (60, 80, and 100 Gy) to establish the optimal dose for further analyses with a larger sample of individuals and to establish in which hours of the day mating occurred with greater frequency. The explanatory variables were the dose applied to the males (0, 60, 80, 100 Gy) and the experimental block, with the dose nested within the latter variable. The rate of mating, the time lapsed until the first mating, and the first mating duration were considered as outcome variables. A glmer analysis [43] (generalized mixed model with random effects; package lme4, in the statistical environment R) was performed on the rate of mating as a variable of response (as an event of success or failure). In the case of the time lapsed until the first mating, the data were treated as the number of observation units until the first mating as well as for the duration of first mating. In the latter case, a GLM model was employed using PASW Statistics 17 [44], and a negative binomial distribution was applied in both cases. Finally, the number of mating events per couple was analyzed as an outcome variable, and in this case, a GLM model with a Poisson distribution was employed. Twenty repetitions were carried out for each dose.

#### 2.5.2. Study of Sexual Behavior at the Optimal Dose

A GLiM marginal model, with repeated observations [45] by the package geepack, version 1.3.12 [46], in the statistical environment R), was used for both choice and no-choice experiments. The GEE (generalized estimating equation) model estimates the population-averaged effects using parameters of a generalized linear model with a possible correlation link between the observations. In this case, due to the different structure of the dataset (fewer repeated measures than in the preliminary analysis and sparse data in the case of mating comparisons), it was preferred to the mixed models, since, in most cases, the female either mated or did not mate with the same male throughout the experiment, leading to the risk of random effect misspecification and potential bias as a consequence [47]. The model design and the autocorrelation structure were chosen based on the optimal parsimony principle using a QIC estimator [48], an equivalent of the Akaike Information Criterion (AIC). All the experiments were repeated temporally in five blocks of about 20 replicates during autumn and early winter to rule out any variability due to the state of rearing or seasonality. Thus, one hundred replicates were performed in both choice and no-choice tests.

No-choice test. The model was applied to three behavioral parameters as variables of response: the rate of mating (mating = 1; no mating = 0), and the female and male rates of feeding (feeding = 1; no feeding = 0). A binomial distribution with a “logit” link was applied to the three outcome variables. Two fixed effects were considered as explanatory variables: the treatment (male irradiated or not) and the day of the experiment, treated as a categorical variable. A full factorial design was chosen for the data regarding mating. In the case of feeding behavior, with the aim to analyze the variations in feeding during the experiment, feeding trends were analyzed separately for males and females as a function of the irradiation treatment. In addition, a matrix structure was assigned to the response variable (feeding of male, feeding of female) to directly compare feeding patterns as a function of the irradiation treatment. Non-linear components of the trends were also tested in the model by the poly function available in R. The comparisons among levels of the categorical variables were performed by the function contrast within the R package contrast, 0.24.2 version.

Choice test. In the choice experiment, the same type of model was adopted, and the same variables of response were considered as in the no-choice test. A matrix structure (mating with the irradiated male, mating with the non-irradiated male) was assigned to the response variable into the binomial model for the two possible alternatives of mating. In this case, a model with the intercept set to zero was performed, and the day of the experiment was considered as the fixed effect. Thus, any deviation from the 50% in the mating proportion (the mean zero in the marginal model) between the two types of males (irradiated or not) was computed within each day. The overall effects were estimated by the ANOVA function in R applied to the binomial model [49]. The same model was applied to the positive (non-irradiated 1/non-irradiated 2) and negative controls (irradiated 1/irradiated 2) for checking the choice experiment. A second GiLM marginal GEE binomial model was used to estimate changes in the overall percentage of time spent in mating considering both the types of males (irradiated 1 + non-irradiated 2; non-irradiated 1 + non-irradiated 2; irradiated 1 + irradiated 2). Two fixed effects were considered: the three treatments (test group, positive control, and negative control) and the effect of the day. On the basis of QIC and significance, the interactions were not considered. In the case of feeding behavior, to directly compare feeding patterns as a function of the irradiation treatment, a matrix structure was assigned to the response variable in the three possible combinations corresponding to three different models: (a) feeding of male 1–feeding of female; (b) feeding of male 2–feeding of female; (c) feeding of male 1–feeding of male 2. With the aim to analyze variations in the feeding trends during the experiment, the days of the experiment and the three groups of treatment (test and controls) were analyzed as fixed effects separately for the two types of males and for the female. The comparisons among levels of the categorical variables were performed by the function contrast within the R package contrast, 0.24.2 version. All the figures of trends and histograms were generated using PASW Statistics 17 [44].

With the purpose of investigating the behavioral patterns under choice conditions, a PCA analysis was conducted (FactoMineR package, version 2.11 [50], in the statistical environment R) on the data recorded for each cage, specifically the total number of hours spent mating or feeding by the 3 individual present in the arena. Five outcome variables were used in the analysis: mating with male “1”, mating with male “2”, feeding of the female, feeding of male “1”, and feeding of male “2”. The types of combinations in the arena (the test group and controls) were considered as a qualitative trait and were excluded from the analysis and used to evaluate the association of this variable with the five behavioral variables. The data were standardized thanks to the PCA function, which, in FactoMineR, processes the data by default to avoid biases in the analysis. The first and second components (PC1/PC2) were used to create a scatter plot with the FactorMineR package.

## 3. Results

### 3.1. Evaluation of Sexual Performance at Three Different Doses

In general, mating occurrence was evenly distributed throughout the day, whereas a decrease in mating activity was observed during the night. A higher rate (%) of mating was observed at the dose of 80 Gy and at 60 Gy in comparison to the non-irradiated control (Table 1), but the values did not reach statistical significance (at 80 Gy: rate of 73.3 ± 3.1 vs. 60.1 ± 3.5 of the non-irradiated; at 60 Gy: 63.5 ± 3.5 vs. 47.5 ± 3.5 of the non-irradiated). An effect of the experimental block was found in the 0–100 Gy group (Table 1), with a lower rate of mating (coef = −4.737 ± 1.936; z = −2.446, 0.0144 *) and a longer time lapsed before the first mating, in comparison to the 0–80 Gy group (coef = 1.271, Wald chisq = 5.904 *p* = 0.015). In any case, a lower rate of mating was found at 100 Gy when compared to the non-irradiated control (14.2 ± 2.5 vs. 21.7 ± 2.9 of the control; coef = −3.674135 ± 1.805881; z = −2.035, *p* = 0.0419 *). As regards the results on the time lapse until the first mating, a time lapse significantly shorter than the control for the dose 80 Gy was recorded (1.21 ± 0.55 observation units vs. 4.67 ± 2.21 of the non-irradiated; coef = 1.269, Wald chisq = 4.237, *p* = 0.039). A shorter time lapse was also observed for the 60 Gy dose in comparison to the non-irradiated control (Table 1), but the difference was not significant. Finally, no appreciable differences were found between the tested doses and the non-irradiated control for the two other parameters, n. of mating events and first mating duration, although a tendency towards a more prolonged time of mating was observed when females mated with irradiated males (at 80 Gy: 57.56 ± 5.64 observation units vs. 43.67 ± 7.65 of the non-irradiated). Considering these results and the results obtained in the experiments on the biological parameters, such as the level of sterility and male longevity [34], the dose of 80 Gy was chosen for further analyses with a wider sampling of individuals. In fact, in a previous work, a sterility of 95% and a life span of 32.5 ± 1.4 days were achieved at 80 Gy, versus the sterility of 90% at 60 Gy and the sterility of 99% obtained at 100 Gy but with a reduced life span (23.8 ± 1.7 days).

### 3.2. Study of Sexual Behavior at the Optimal Dose

#### 3.2.1. No-Choice Test

No detrimental effects on mating were observed in the group with irradiated males. The percentages of cases with any mating in the cage throughout the whole experiment were totally equivalent between the irradiated and non-irradiated male groups (25% of cases without mating events in the cages with irradiated males and 26% of cases within the cages with non-irradiated males). On the contrary, the irradiated males at the dose of 80 Gy were able to mate with the females as well as or even better than the non-irradiated individuals. In fact (Table 2), excluding the cages with zero mating, females mated with irradiated males for 40.1 ± 1.1% of the time, in comparison to 30.5 ± 1.1% of the time when the normal males were confined with females. This gap was statistically significant (Table 3), according to the GiLM GEE model (effect of the irradiated in comparison to the non-irradiated: Wald = 10.12 *p* = 0.00146). Observing the trend in mating rate, the mating with the irradiated male started at much higher levels than in the cross with the non-irradiated (38.5 ± 1.9% vs. 21.5 ± 1.5%; contrasts: Z = 3.18, *p* = 0.0015), continuing to increase on the second day (day 1–day 2 contrasts: Z = 2.26, *p* = 0.0236) and remaining significantly higher than in the cross with the non-irradiated male (47.5 ± 1.9% vs. 31.1 ± 1.8% of the non-irradiated; contrasts: Z = 2.18, *p* = 0.0083). However, mating activity decreased on the third day (day 2–day 3 contrasts: Z = 3.48, *p* ≤ 0.001), with values similar to the non-irradiated group (37.0 ± 1.9% vs. the 39.2 ± 1.9% of mating rate recorded on non-irradiated males). In the case of mating with the non-irradiated males, on the first day, mating occurrence started at low levels (21.5 ± 1.5%) and then showed a positive trend until the end of the experiment (interaction non-irradiated*day 3: Wald = 11.26, *p* = 0.00079). Thus, the irradiation treatment increased mating chances until the second day of the experiment, then becoming almost equivalent to the cross with the non-irradiated on the last day (Figure 1, Table 2). As regards feeding behavior, no significant effects of irradiation were detected, neither when comparing irradiated males to females, nor when comparing irradiated males among themselves (Table A3). The only robust effect was the higher occurrence of female feeding in comparison to males (intercept: Wald = 22.24, *p* = 2.4 × 10^−6^), regardless of irradiation treatment (Table 2), although on the third day, the gap between males and females significantly decreased (day 3: coef. = −0.0466, Wald = 5.05, *p* = 0.025). Instead, a common feature was shared with the two males and the females: a sinusoidal trend emerged after a steady increase in feeding activity on the first day (Figure 2). This trend was verified by a GiLM GEE model applying a non-linear cubic trend (Appendix A, Table 1 and Table 2, and Figure A4).

#### 3.2.2. Choice Test

No significant effects in bagrada behavior due to the marking with colors were observed. In general, the singular events of mating were often very long, in several cases covering the entire time of the experiment (21 h). No relevant differences were recorded in the number of cages without any mating throughout the experiment (21% on the test non-irradiated–irradiated, 15% in the positive control, and 14% in the negative control). Once mated, females rarely changed their partner during the test. In fact, when the total time spent in mating was compared, a negative correlation was found between the mating occurrence with males “1” and “2” (Spearman correlation coefficient = −0.608 ***). This behavioral trait was confirmed in the PCA analysis, where the two males were found strongly anti-correlated, regardless of the type of test (0.83 and −0.79 on the second component of variability; see Figure A1 in the Appendix A). For the same reason, many males did not mate at all during the choice experiment. Considering only the cages where mating occurred, in the test irradiated–non-irradiated, 30% of the irradiated males never mated, and 41% of non-irradiated males never mated. These values were similar in both the controls, with values of 30% and 41% for the two non-irradiated males and 40% and 41% for the two irradiated males. Conversely, feeding activity was generally shared among the individuals (according to the PCA analysis on the first component of variability, 0.88, 0.85, and 0.71, respectively, for the female, male “1”, and male “2”), where mainly the males followed the feeding behavior of the female (r = 0.617 *** for male “1” and r = 0.559 *** for male “2”). No relevant differences in the basic behavioral pattern were found among the three tested experimental schemes (non-irradiated–irradiated; non-irradiated–non-irradiated; irradiated–irradiated). In any case, as reported in Table 4, mating occurrence with the irradiated male was significantly higher than with the non-irradiated male. In fact, the percentage of mating occurrence was always higher in the case of the irradiated male during the 3 days of the experiment. As Table 5 shows, the overall effect of irradiation on mating was significant (Wald = 9.17, *p* = 0.027). Particularly, on the first day, the mating with the treated male was almost three times more frequent compared to that with the non-irradiated male (17.4 ± 1.6% vs. 6.3 ± 1.0%; Wald = 7.40, *p* = 0.0065). The percentage of the time spent in mating was more than double on the second day (35.6 ± 2.0% vs. 16.6 ± 1.6%; Wald = 7.23, *p* = 0.0072), remaining higher also on the third day, even though the gap was slightly reduced (43.6 ± 1.2% vs. 29.1 ± 1.9%; Wald = 2.59, *p* = 0.1077). The effect of males “1” and “2” in the positive and negative controls (Table 5) was not significant in any case, as expected (Table 5).

The effects, as a relative proportion between irradiated and non-irradiated males, are reported in Figure 3. Considering the overall mean, the mating proportion was 64.8%, reaching the highest value of 72.2% on the first day, and then slightly decreasing on the second and third days (68.2% and 59.9%, respectively). The rate of mating increased with time (Figure 4), but no appreciable differences were found in the trend pattern among the three treatments, since no significant interactions were found. The GiLM GEE results in Table A4 show the effects of the day and of the treatments (test experiment, positive control, and negative control) on overall mating occurrence, calculated as the sum of the percentage of time spent in mating with both males. Mating occurrence increased with days, which were considered as a main effect in the model (day 2: coef. = 1.094, Wald = 78.71, *p* < 2 × 10^−16^; day 3: coef. = 1.819, Wald = 118.85, *p* < 2 × 10^−16^), since no interactions with the type of cross were found. Interestingly (Table 4), a significantly higher mating activity was found when the two irradiated males were introduced in the cage with the female in comparison with the other two crosses (coef. = 1.194, Wald = 17.71, *p* = 0.000026). In addition, the percentage of the time spent in mating with the irradiated male in the test experiment irradiated/non-irradiated (Table 4) was very close to the percentage recorded for each male in the irradiated/irradiated control (32.2 ± 1.1% for the irradiated male in the test experiment vs. 34.4 ± 1.1% for male “1” and 33.5 ± 1.1% for male “2” in the irradiated/irradiated negative control). On the contrary (Table 4), the non-irradiated males mated more in the non-irradiated/non-irradiated positive control than in the test experiment irradiated/non-irradiated (25.6 ± 1.0% vs. 17.4 ± 0.9%), confirming that mating activity was partially inhibited in the competition with the irradiated male. Figure A1 summarizes the overall results concerning mating in the choice experiments, presenting the total number of hours spent on mating.

As regards the feeding behavior, no significant differences in the percentage of time spent in feeding were found between the irradiated and non-irradiated males throughout the test experiment, as the GiLM GEE model shows (Table 6). The overall means were very similar (Table 6) with 43.1 ± 1.1% of the time for the irradiated male and 41.2 ± 1.1% for the non-irradiated male. As in the case of the no-choice test, an increasing trend with an apparent sinusoidal component was observed during the experiment, and this effect was shared among the three individuals confined into the cages (Figure A3). As already observed in the no-choice experiment, the female fed much more than the two males (feeding occurrence in the test experiment: 61.6 ± 1.2% vs. 43.1 ± 1.2% of the irradiated male and 41.2 ± 1.2% of the non-irradiated male), regardless of the treatment applied (Table 6). According to the GiLM GEE model of Table A5, this difference was highly significant for both the comparisons with males (overall effect: X^2^ = 54.8, *p* = 5.7 × 10^−14^ in the female–non-irradiated male comparison; X^2^ = 62.5, *p* = 1.8 × 10^−13^ for the female–irradiated male comparison). Finally, the overall feeding occurrence was higher in the irradiated/non-irradiated group, compared to the other two control groups (Table 6).

## 4. Discussion

In the first experiment, results showed that at doses of 60 and 80 Gy, irradiated males maintained a mating frequency comparable to that of non-irradiated males. Specifically, at 80 Gy, the time interval before first mating was significantly shorter than that in the control, while at 60 Gy, this difference did not reach statistical significance. This suggests that irradiated males at these doses maintain a good reproductive capacity, comparable to that of non-irradiated males. However, at the highest dose (100 Gy), a significant decrease in mating frequency was observed with an increase in the time required for the first mating. Among the doses tested, 80 Gy proved to be the most promising, balancing a high level of induced sterility (95%), preserving longevity [34], and sexual performance comparable to the control insects. Although 60 Gy is an efficient treatment (90% sterility) [34], the greater competitiveness recorded at 80 Gy (reduced time before first mating and longer duration of initial copulation) makes the latter preferable for SIT purposes. In fact, simulating a field release, the long duration of the first copulation, associated with timely mating with wild females, could increase the success of SIT. At 100 Gy, while achieving complete infertility, males experience negative effects on their competitiveness. This reduction in sexual performance may be explained by alterations in the intraspecific vibrational signals emitted by irradiated males, as shown in previous studies on *B. hilaris*. At this dose, the males emitted signals with lower peak frequencies and showed reduced mating ability [51]. A further step will be to examine the influence of irradiation on pheromone production, as volatile and contact compounds play a crucial role in sexual communication in this species [42].

The second experiment confirmed and extended these findings by examining the competitiveness of males irradiated at 80 Gy in no-choice and choice tests over three days of observation (Figure 5).

Under choice conditions, females showed a preference for irradiated males, who spent more time mating than non-irradiated males (32.2 ± 1.1% vs. 17.4 ± 0.9%). In this case, competition between males seems to incentivize performance; irradiated males have increasing mating rates from day 1 to day 3 (17.4 ± 1.6%; 35.6 ± 2.0%; 43.6 ± 2.1%, respectively), in contrast to the decline under no-choice conditions. The second competing male was often found near or above the copulating pair, waiting for an opportunity to mate with the female (personal observation). Thus, the duration of copulation for bagrada may be influenced by the social environment [52] as a means of protecting females from other males. A form of mate guarding is present in this species [8], as confirmed by two behavioral traits. The first is the prolonged duration of copulation [8], which is longer than that required for the transmission of seminal fluids and spermatozoa and is employed in order to protect females from other inseminations [53,54]. The second trait concerns the sperm competition mechanism, in particular, the mixing sperm observed in *B. hilaris* [55], which occurs from multiple matings within the spermatheca [56]. In this way, all sperm have an equal chance of fertilization (“the fair raffle game”) [57], and with prolonged copulas, males prevent subsequent insemination by another male, thus ensuring that their sperm will be used for the next ovipositional event [58].

An interesting finding emerged from the negative control (female with two irradiated males), where a mating rate of 67.9 ± 1.1% was found compared to 50.8 ± 1.2% for the positive control (female with two non-irradiated males). This observation supports the hypothesis that the competitiveness of irradiated males may, under the pressure of active males, be equal or superior to non-irradiated males. This phenomenon might be explained by the fact that irradiation can induce males to exhibit more frequent abdominal contractions during mating, with a significantly higher rate of contractions per minute than in untreated males [51]. Such hyperkinetic behavior could also represent one of the reasons why females sometimes prefer the irradiated male under conditions of choice.

It should be noted that long copulation times reduced the number of males actually mating, as many females did not change partners within the three-day period. In *Nezara viridula*, virgin females have longer copulation times than when mating twice [52]. In our study, a similar behavior was observed in *B. hilaris*.

The results obtained regarding feeding activity confirmed the observations of Huang and colleagues [59]: females fed longer than males, especially in the first two days, which could be due to a higher demand for plant nutrients useful for vitellogenesis. In addition, no differences in feeding behavior were observed between non-irradiated and irradiated males, both showing it mainly during copulation.

This result is promising for the application of SIT; if insect damage is mainly caused by females, the release of males only could allow the use of SIT with little or no negative consequences on host plants [60].

Another promising result is the maintenance of the performance of the irradiated male over the three days following irradiation. From a release perspective, it is critical that the male remains competitive over time. Future tests should aim to monitor sexual performance over a longer period, as irradiation may have a negative influence over time [61].

Overall, the data indicate that 80 Gy is an effective dose of irradiation, inducing 95% sterility and maintaining excellent or even greater sexual performance than that of non-irradiated males. It was found that there were no mating preferences between non-irradiated and irradiated males, which could have reduced the effectiveness of the technique [62]. These observations raise interesting questions regarding the dynamics of sexual selection and the biological consequences of irradiation. Irradiation treatment can affect the mating behavior of insects [63]. In the case of *B. hilaris*, the enhanced sexual performance observed in irradiated males may stem from a prolonged mating duration, which likely maximizes sperm transfer and compensates for reduced fertility. Because females do not receive any immediate indication of male infertility during copulation, a male displaying adequate courtship behavior will still be accepted.

Within a polyandrous context, sterile *B. hilaris* males must compete with their non-irradiated counterparts and may thus invest additional resources in courtship and prolonged copulation. This could reflect a mechanism of hormesis, that is, an over-compensatory response triggered by mild stress [64]. Hormesis is widely recognized as a general adaptive phenomenon; when homeostasis is disrupted and minor damage occurs, organisms typically repair the damage and overshoot baseline levels to ensure effective recovery [64,65]. In insects, this overcompensation often manifests as an increase in reproduction, but it can also bolster longevity, growth, behavior, and immunity [65].

Furthermore, sexual communication in *B. hilaris* includes vibrational signals emitted by males. Irradiation alters the frequency and intensity of these signals [51], which paradoxically stimulates females and further enhances the reproductive success of sterilized males.

This result complements similar observations in other species, as no significant changes were observed for *Anastrepha fraterculus* Wiedemann (Diptera: Tephritidae) treated at doses of 40, 70, and 100 Gy compared with the control [66], as well as for *Drosophila suzukii* Matsumura (Diptera: Drosophilidae), in which no impact on the competitiveness of males irradiated at 120 and 200 Gy was observed [67,68].

In general, the irradiation process may affect a number of quality parameters, including longevity [69], mating competitiveness, sperm transfer [70], flight ability [71], production, and response to pheromones [72]. In *B. hilaris*, a reduction in longevity following irradiation has not been detected [34]; and this study found that at moderate doses (60 and 80 Gy), the competitiveness of the irradiated male was not altered. Regarding the study of sperm competition in *B. hilaris*, if the non-irradiated virgin female is exposed first to an irradiated male and then to an non-irradiated male, complete recovery of fertility does not occur [55].

In the current study, the experiments were conducted in the laboratory under controlled conditions, and the results must be validated in semi-field and field tests to prove the real value of SIT for controlling bagrada. Mass rearing over multiple generations can result in challenges such as inbreeding and adaptation to laboratory conditions, making it difficult to maintain the biological quality and competitiveness of irradiated males for a successful SIT program [73]. Future research will focus on optimizing colony management strategies to preserve the fitness of mass-reared *B. hilaris*. In addition, to determine an effective application protocol, it will be necessary to conduct population dynamics experiments and identify the overflooding ratio (i.e., the ratio of irradiated to non-irradiated males required for field release to achieve a fertility suppression effect on the population).

Consequently, given the information obtained thus far on *B. hilaris*, the application of SIT in an area-wide IPM context gains credibility and could be now considered. Indeed, in addition to the quality parameters mentioned above, the success of this technique could be enhanced if applied in geographically and ecologically well-defined areas, such as the island of Pantelleria (Italy), in combination with classical biological control.

## 5. Conclusions

The present study found that irradiation doses of 60 and 80 Gy had no negative effect on the mating ability of *B. hilaris* males, in contrast to the 100 Gy dose, which resulted in reduced sexual performance. This allowed the intermediate dose (80 Gy) to be identified as a possible value for the application of SIT in bagrada due to the maintenance of sexual performance despite induced high sterility (95%). In fact, under both choice and no-choice conditions, the males irradiated at 80 Gy spent more time mating than the non-irradiated males, and in most cases were preferred by the females. In addition, the lower feeding activity of males compared to females suggests that the field release of males could only ensure minimal impact on the cropping system. In order to assess the feasibility of this technique, it will also be important to investigate the overflooding ratio. Field release of only irradiated males, combined with an integrated management approach, could ensure a minimal impact on crops and support sustainable strategies to contain this invasive species.

## Figures and Tables

**Figure 1 insects-16-00391-f001:**
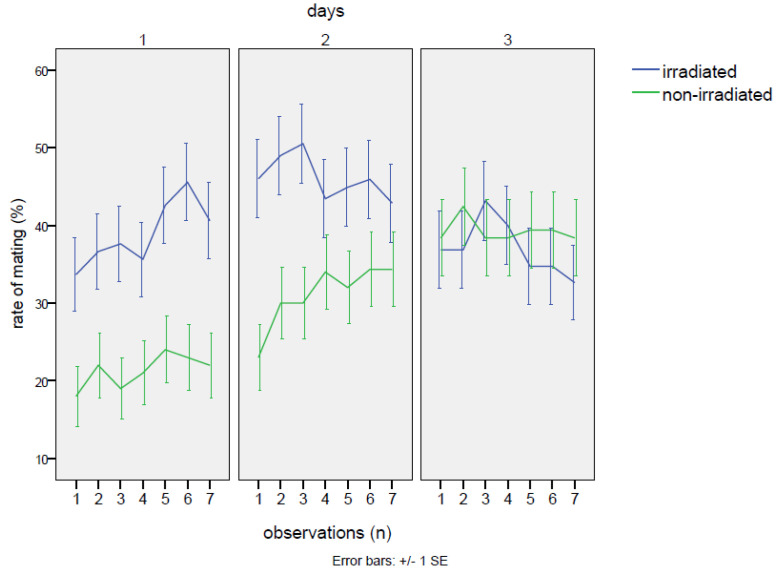
No-choice test: mating trends measured as the percentage of time spent in mating during the three days of the experiment in the two crosses with the irradiated male and the non-irradiated male of *Bagrada hilaris.* The blue line represents the irradiated male, and the green line the unirradiated male.

**Figure 2 insects-16-00391-f002:**
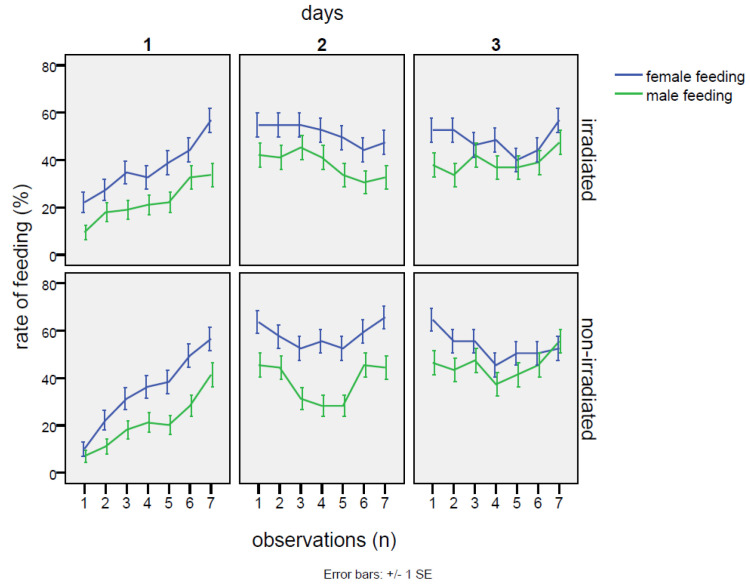
No-choice test: feeding trends measured as the percentage of time spent in feeding during the three days of the experiment in the two crosses with the irradiated male and non-irradiated male of *Bagrada hilaris*. The blue lines indicate the feeding activity of the female, the green lines indicate that of the male (graph above: irradiated male; graph below: unirradiated male), under no-choice conditions.

**Figure 3 insects-16-00391-f003:**
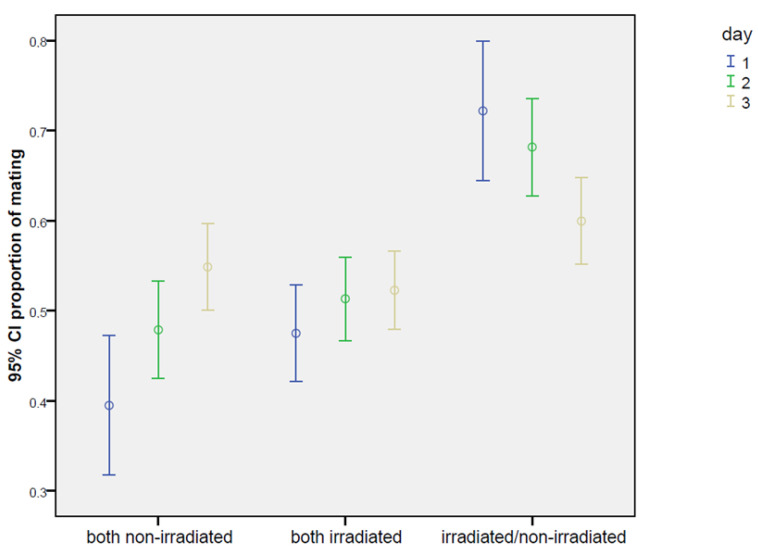
Choice test: ratio between the two types of mating events (irradiated/non-irradiated male), under choice conditions (one non-irradiated female confined with one unirradiated male and one irradiated male) in *Bagrada hilaris*. The test experiment is also compared with the positive (one female with two non-irradiated males) and negative controls (one female with two irradiated males). Blue lines refer to the first day of observation, green to the second day, and yellow to the third day.

**Figure 4 insects-16-00391-f004:**
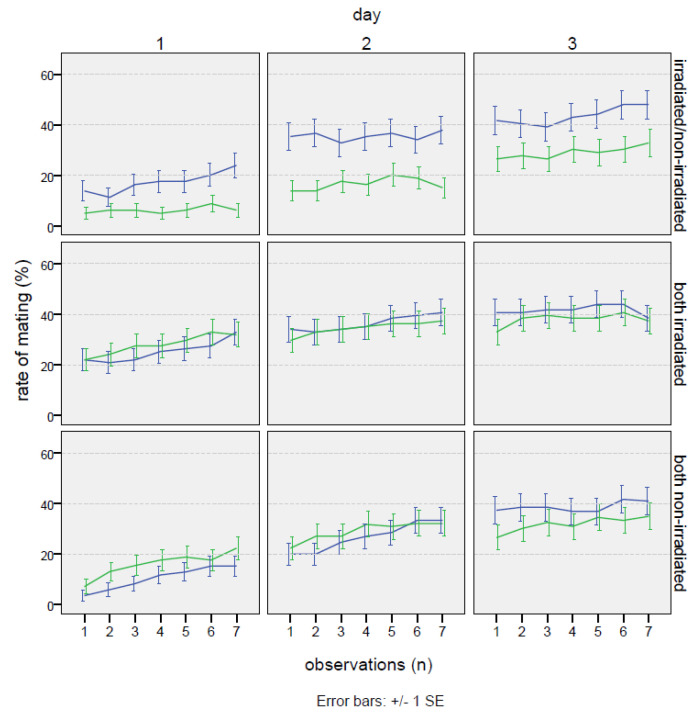
Choice test: mating occurrence trends expressed as percentages of time, under choice conditions (one non-irradiated female with one non-irradiated male and one irradiated male) in *Bagrada hilaris*. The blue lines represent the irradiated male (or male 1 in the positive and negative controls), and the green lines the non-irradiated one (or male 2 in the positive and negative controls).

**Figure 5 insects-16-00391-f005:**
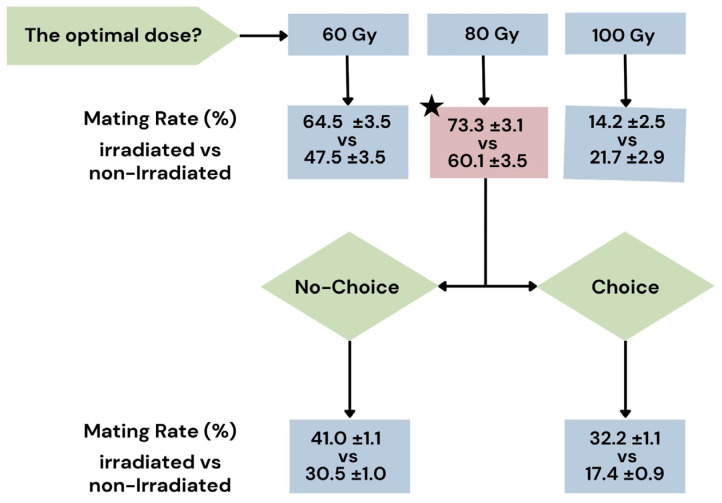
Flow chart illustrating the results of the two experiments. In the first experiment, the optimal irradiation dose was determined by testing 60, 80, and 100 Gy, revealing that an 80 Gy dose produced the highest mating rate in *Bagrada hilaris*. Subsequently, tests were conducted under choice (one non-irradiated female, one irradiated male, and one non-irradiated male) and no-choice (one non-irradiated female and one irradiated male) conditions. The black pentagram indicates the dose considered optimal, which is 80 Gy.

**Table 1 insects-16-00391-t001:** Means and standard errors of the analyzed response variables rate of mating, time lapse before mating, duration of first mating, and mating events per couple of *Bagrada hilaris*, as a function of the irradiation doses and of the experimental block. The pairs consisted of one non-irradiated virgin female and one irradiated male (at 60, 80, and 100 Gy). The corresponding controls consisted of one non-irradiated virgin female and one unirradiated male. Values significantly different from the control are followed by asterisks, according to the applied model.

Experimental Block	Dose (Gy)	Mating Rate (%)	Time Lapse Before First Mating (obs. n.)	First Mating Duration (obs. n.)	N. of Mating Events (n.)
1	0	47.5 ± 3.5	9.33 ± 4.09	46.20 ± 6.24	1.25 ± 0.22
	60	64.5 ± 3.5	7.06 ± 3.77	49.94 ± 4.78	1.40 ± 0.24
2	0	60.1 ± 3.5	4.67 ± 2.21	43.67 ± 7.65	1.10 ± 0.19
	80	73.3 ± 3.1	1.31 * ± 0.55	57.56 ± 5.64	1.16 ± 0.21
3	0	21.7 ± 2.9	16.64 ± 4.40	28.55 ± 7.78	0.55 ± 0.11
	100	14.2 * ± 2.5	26.00 ± 10.59	34.33 ± 8.80	0.30 ± 0.11

**Table 2 insects-16-00391-t002:** No-choice test: mean percentage of mating and feeding on the first, second, and third days of the experiment by *Bagrada hilaris* under no-choice conditions, i.e., one unirradiated female with one irradiated male (80 Gy). The effects of male irradiation were compared with the cages with non-irradiated males for these biological parameters. Values followed by asterisks are significantly different at *p* = 0.05, as calculated by contrast analysis within the GiLM GEE model. In the case of mating, the asterisk refers to the statistical significance of comparison between the irradiated and non-irradiated groups within the day. In the case of feeding outcomes, the asterisks refer to the comparisons between the females and the males (no significant differences were found between the two treatments for both males and females within each day). Levels of significance are reported in the table according to the conventional notation by asterisks: no symbols, *p* > 0.05; *p* ≤ 0.01 **; *p* ≤ 0.001 ***.

	Male Treatment		Mating		Female Feeding		Male Feeding	
Days		N Cages	% of Time	±s.e	% of Time	±s.e.	% of Time	±s.e.
1	irradiated	95	38.5	±1.9 **	36.7	±1.9 ***	22.3	±1.6
	non-irradiated	99	21.5	±1.5	34.9	±1.8 ***	21.1	±1.6
2	irradiated	95	47.5	±1.9 **	51.1	±1.9 ***	38.0	±1.9
	non-irradiated	99	31.1	±1.8	58.2	±1.9 ***	38.2	±1.8
3	irradiated	95	37.0	±1.9	48.7	±1.9 **	39.1	±1.9
	non-irradiated	99	39.2	±1.9	53.5	±1.9 **	45.3	±1.9
Total mean	irradiated	95	41.0	±1.1 **	45.5	±1.1 ***	33.1	±1.1
	non-irradiated	99	30.5	±1.0	49.9	±1.1 ***	34.9	±1.0

**Table 3 insects-16-00391-t003:** No-choice test: GiLM marginal GEE model, under no-choice conditions, i.e., one unirradiated female with one irradiated male (80 Gy). Mating rate is the variable of response. Data were analyzed in the binomial form (0 = no mating; 1 = mating). Two fixed factors were considered in the analysis: the effect of treatment (irradiated or not) and the day of the experiment. A full factorial design was applied. The “non-irradiated” and the “day 1” levels were fixed as reference levels and thus are not reported in the table. The ANOVA table for the estimation of the overall effects is also reported below. Levels of significance are reported in the table according to the conventional notation by asterisks: no symbols, *p* > 0.05; *p* ≤ 0.05 *; *p* ≤ 0.01 **; *p* ≤ 0.001 ***.

Fixed Effects	Estimate	±s.e.	Wald	*p*-Value
(Intercept)	−0.4416	0.1698	6.76	0.00930 **
Irradiated	0.8662	0.2722	10.12	0.00146 **
Day 2	0.3526	0.1558	5.12	0.02361 *
Day 3	−0.0674	0.1734	0.15	0.69775
Irradiated × day 2	−0.1559	0.2115	0.54	0.46095
Irradiated × day 3	−0.9271	0.2763	11.26	0.00079 ***
Analysis of “Wald statistic” Table				
	df	X^2^	*p*-value	
Treatment	1	4.88	0.02714 *	
Days	2	15.95	0.00034 ***	
Treatment × days	2	14.77	0.00062 ***	

**Table 4 insects-16-00391-t004:** Choice test: mean percentage of mating occurrence on the first, second, and third days of the experiment. The test experiment with the irradiated male and the non-irradiated male in the cage is also compared with the positive control (both males non-irradiated) and negative control (both males irradiated). In the test experiment, male “1” is the irradiated one. Total mean values are also shown. Values followed by different letters are significantly different at *p* = 0.05, as calculated by contrast analysis within the GiLM GEE model. Comparisons are made among the days of the experiment within each treatment. Capital letters refer to the overall comparisons among treatments. Significant differences between males “1” and “2” are marked by asterisks. Levels of significance are reported in the table according to the conventional notation by asterisks: no symbols, *p* > 0.05; *p* ≤ 0.05 *; *p* ≤ 0.01 **.

Male Treatment	Day	Cages (*n*)	Total Rate of Mating (%)	±s.e.	Rate of Mating (Male 1 *) (%)	±s.e.	Rate of Mating (Male 2 *) (%)	±s.e.
Irradiated/ non-irradiated * male 1 = irradiated * male 2 = non-irradiated	1		23.7 a	1.8	17.4 a	1.6 **	6.3 a	1.0
	2	79	52.3 b	2.1	35.6 b	2.0 **	16.6 b	1.6
	3		72.7 c	1.9	43.6 b	2.1	29.1 c	1.9
	Total mean		49.6 A	1.2	32.2 A	1.1 *	17.4 A	0.9
Both non-irradiated	1		26.5 a	1.8	10.4 a	1.3	16.0 a	1.5
	2	85	55.7 b	2.0	26.7 b	1.8	29.1 b	1.9
	3		70.5 c	1.9	38.7 c	2.0	31.8 b	1.9
	Total mean		50.8 A	1.2	25.2 A	1.0	25.6 B	1.0
Both irradiated	1		52.0 a	2.0	25.3 a	1.7	27.8 a	1.8
	2	91	70.8 b	1.8	36.4 b	1.9	34.5 ab	1.9
	3		80.0 b	1.6	41.8 b	2.0	38.1 b	1.9
	Total mean		67.9 B	1.1	34.4 A	1.1	33.5 B	1.1

**Table 5 insects-16-00391-t005:** Choice test: GiLM GEE binomial model, under choice conditions (one non-irradiated female confined with one non-irradiated male and one irradiated male). Mating occurrence is the variable of response. The day of the experiment is the fixed effect. The effect of the treatment (irradiated or not in the test experiment, or male 1 and male 2 in the positive (one female with two non-irradiated males) and negative control (one female with two irradiated males) is estimated within each day. The overall effect is estimated by the ANOVA function for binomial data. Levels of significance are reported in the table according to the conventional notation by asterisks: no symbols, *p* > 0.05; *p* ≤ 0.01 **.

Male Treatment	Fixed Effects	Estimate	±s.e.	Wald	*p*-Value
	day 1	0.2979	0.1096	7.40	0.0065 **
Irradiated/non-irradiated	day 2	0.2434	0.0906	7.23	0.0072 **
	day 3	0.1329	0.0826	2.59	0.1077
	Test ANOVA	df	X^2^	*p*-value	
		3	9.17	0.027	
	Fixed effects	estimate	±s.e.	Wald	*p*-value
Both non-irradiated	day 1	−0.0283	0.0891	0.10	0.75
	day 2	−0.1404	0.1207	1.35	0.24
	day 3	0.0647	0.0825	0.62	0.43
	Test ANOVA	df	X^2^	*p*-value	
		3	4.23	0.24	
	Fixed effects	estimate	±s.e.	Wald	*p*-value
Both irradiated	day 1	−0.0423	0.0911	0.22	0.64
	day 2	0.0178	0.0802	0.05	0.82
	day 3	0.0339	0.0747	0.21	0.65
	Test ANOVA	df	X^2^	*p*-value	
		3	1.28	0.73	

**Table 6 insects-16-00391-t006:** Choice test: mean percentage of feeding occurrence on the first, second, and third days of the experiment. Feeding activity between females, irradiated males, and non-irradiated males are compared with the positive and negative controls for this biological parameter. Total mean values are also shown. Values followed by different letters are significantly different at *p* = 0.05, as calculated by contrast analysis within the GiLM GEE model. Comparisons are made among the days of the experiment within each treatment. Capital letters refer to the overall comparisons among treatments. Male 1 and 2 indicate the two males in the arena. In the test experiment the male 1 is the irradiated male while male 2 is the untreated as reported by the asterisks.

Male Treatment	Day	Cages (*n*)	Feeding Rate of Female (%)	±s.e.	Feeding Rate of Male 1 * (%)	±s.e.	Feeding Rate of Male 2 * (%)	±s.e.
Irradiated/ non-irradiated * male 1 = irradiated * male 2 = non-irradiated	1		57.7 a	2.1	34.4 a	2.0	30.6 a	2.0
	2	79	61.5 a	2.1	43.6 ab	2.1	41.2 ab	2.1
	3		65.6 a	2.0	51.4 b	2.1	51.9 b	2.1
	Total mean		61.6 A	1.2	43.1 A	1.2	41.2 A	1.2
Both non-irradiated	1		28.5 a	1.9	15.9 a	1.5	16.6 a	1.5
	2	85	58.8 b	2.0	34.8 b	2.0	40.3 b	2.0
	3		53.8 b	2.1	38.0 b	2.0	34.1 b	2.0
	Total mean		47.1 B	1.2	29.5 B	1.1	30.3 B	1.1
Both irradiated	1		30.1 a	1.8	15.9 a	1.4	19.5 a	1.6
	2	91	54.5 b	2.0	38.8 b	1.9	37.4 b	1.9
	3		51.6 b	2.0	33.3 b	1.9	34.7 b	1.9
	Total mean		45.4 B	1.1	29.3 B	1.0	30.5 B	1.1

## Data Availability

The original contributions presented in the study are included in the article, further inquiries can be directed to the corresponding authors.

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
