# Peer review of "Sterile but Sexy: Assessing the Mating Competitiveness of Irradiated Bagrada hilaris Males for the Development of a Sterile Insect Technique"

_insects, 2025, doi:10.3390/insects16040391_

Round 1
Reviewer 1 Report
Comments and Suggestions for Authors
Dear authors,
The manuscript "Sterile but sexy: assessing the mating competitiveness of irradiated Bagrada hilaris males for the development of a Sterile Insect Technique" is unpublished and brings very important information that will enable the application of the sterile insect technique (SIT) for the control of B. hilares, a species of stink bug that causes great damage to agriculture.
The introduction and objectives are information-rich and clear.
The methodology is well described, many analyses were performed and the data was extremely useful. The authors say they used newly emerged, virgin adults, but there is no comment on the peak period of sexual maturation and copulation. It is also unclear how the sterility of irradiated males was determined.
The results were quite detailed, but the number of figures and tables could perhaps be reduced by merging information.
The discussion begins with a flowchart summarizing the results obtained, which was a fantastic idea.
However, some information could be removed. For example, the last sentence of the title of Figure 5, lines 564 and 565, which could be written in the text of discussion.
Furthermore, lines 566 to 579 and then 581 and 582 are information that is already in the objectives and results and could be removed.
It was mentioned (Introduction) that the use of SIT to control B. hilares would be viable in greenhouses, as it would be possible to eradicate them, but it should be considered that the damage to the plants, caused by feeding, could be as severe as in open fields and there could also be re-infestation from the fields. To solve this, perhaps one could consider inducing a mutation in this species that affects the males' mouthparts, making them unable to feed. In this case, they could be overfed before irradiation, so that they would be able to survive a few days without food to mate with females in the field.
Mating results were surprisingly better for irradiated males. However, most of the insects used in the experiments were from the field and raised for a few generations in the laboratory. Therefore, one must bear in mind the negative factors that mass breeding brings over generations, in the case of a biofactory for breeding sterile males, which can negatively affect the performance of sterile males with wild females present in the field.
Reviewer 2 Report
Comments and Suggestions for Authors
Manuscript ID: insects-3521866-peer-review-v1
Title: Sterile but sexy: assessing the mating competitiveness of irradiated Bagrada hilarismales for the development of a Sterile Insect Technique
In this study, the effects of irradiation on the sexual performance and selection of painted bug Bagrada hilaris were evaluated. Results could provide essential information for controlling B. hilaris using SIT.
This study appears to have been well conceived and executed. Insects should be an appropriate outlet. However, I think there are some significant flaws that need to be addressed before it can be published.
Major comments
- Why are there only three dose gradients instead of more? My confusion comes from the fact that on line 187, the author cited reference [40], while on line 197, the author cited reference [34]. Therefore, I am unclear about the basis for the author's dosage setup.
- In the experiment of 2.3. Evaluation of sexual performance at three different doses, which indexes were used to test the sexual performance? Authors needs to clarify it clearly.
- To investigate the impact of irradiation on sexual selection, the authors conducted a series of experiments. However, it is unclear why the authors did not examine the effects of irradiated females on sexual selection when paired with non-irradiated males.
- In table 1, there is no statistical analysis was conducted among the three treatments: 60, 80, and 100 Gy. Therefore, I am uncertain why the author reached this conclusion, i.e., “Observational tests showed that the doses of 60 Gy and 80 Gy showed no difference in mating times compared to non-irradiated males, in contrast to 100 Gy. Thus, 80 Gy was identified as the most promising dose” in the abstract (Line 37-39). Furthermore, why was 60 Gy identified as the most promising dose when there was no difference in mating times between 60 Gy and 80 Gy?
- Why irradiated males appear to perform better than non-irradiated males. Please interpret it in the discussion.
- The study’s results showed that females preferred these irradiated males. Please clarify the potential causes, for example in lines 622-624.
- The development of a SIT based on the current findings is hindered by the absence of semi-field and field test results. Additionally, it is essential to explore potential application scenarios and address the challenges that must be overcome to achieve successful suppression of the hilaris population using SIT.
Minor comments
- It is recommended that Tables 3-4 be removed, and the corresponding results should be incorporated within the main text.
- It is recommended that Figure 5 be removed.
- Line 599-602, they are repeated with results, which should be deleted.
Round 2
Reviewer 2 Report
Comments and Suggestions for Authors
I am satistified with the revision.